# The Role of Non-Coding RNAs in MYC-Mediated Metabolic Regulation: Feedback Loops and Interactions

**DOI:** 10.3390/ncrna11020027

**Published:** 2025-03-18

**Authors:** Aliaa Amr Alamoudi

**Affiliations:** 1Department of Clinical Biochemistry, Faculty of Medicine, King Abdulaziz University, Jeddah 21589, Saudi Arabia; aaalamoudi2@kau.edu.sa; 2Regenerative Medicine Unit, King Fahd Medical Research Center, King Abdulaziz University, Jeddah 21589, Saudi Arabia

**Keywords:** non-coding RNA, MYC, glycolysis, Warburg-effect, tumor-metabolism, long non-coding RNA, circular RNA, microRNA

## Abstract

Metabolic reprogramming is a hallmark of cancer, crucial for supporting the rapid energy demands of tumor cells. MYC, often deregulated and overexpressed, is a key driver of this shift, promoting the Warburg effect by enhancing glycolysis. However, there remains a gap in understanding the mechanisms and factors influencing MYC’s metabolic roles. Recently, non-coding RNAs (ncRNAs) have emerged as important modulators of MYC functions. This review focuses on ncRNAs that regulate MYC-driven metabolism, particularly the Warburg effect. The review categorizes these ncRNAs into three main groups based on their interaction with MYC and examines the mechanisms behind these interactions. Additionally, we explore how different types of ncRNAs may collaborate or influence each other’s roles in MYC regulation and metabolic function, aiming to identify biomarkers and synthetic lethality targets to disrupt MYC-driven metabolic reprogramming in cancer. Finaly, the review highlights the clinical implications of these ncRNAs, providing an up-to-date summary of their potential roles in cancer prognosis and therapy. With the recent advances in MYC-targeted therapy reaching clinical trials, the exciting potential of combining these therapies with ncRNA-based strategies holds great promise for enhancing treatment efficacy.

## 1. Introduction

It is known that metabolic reprogramming is a tumor hallmark that is required to support the rapid energy needs and biosynthetic demands of growing tumors [1,2].

MYC is a key transcription factor that governs numerous cellular processes, including metabolism, growth, and differentiation [3,4,5,6]. In cancer, MYC is often deregulated and overexpressed, acting as a master regulator of metabolic reprogramming that sustains tumor survival and proliferation. A key hallmark of MYC’s metabolic function is its pivotal role in driving the Warburg effect by enhancing glycolysis, where it increases the expression of glucose transporters and enzymes involved in the glycolytic pathway, thus accelerating the conversion of glucose to lactate even in the presence of oxygen [7,8,9].

Despite the wealth of reviews addressing MYC’s metabolic functions [7,8,9], there remains a gap in comprehensive analyses that describes the mechanisms and factors influencing MYC’s metabolic roles. A deeper understanding of these mechanisms is crucial for mapping how various factors impact MYC’s activity and metabolic regulation, thus uncovering potential molecular targets that regulate or modulate MYC’s metabolic function.

A key area that has emerged is the role of non-coding RNAs (ncRNAs), which have been shown to modulate MYC’s functions, including metabolic functions, through a variety of mechanisms. With the recent advancements in RNA-based therapies—such as RNA interference, small molecules, and antisense oligonucleotides—which have demonstrated the potential to modulate ncRNA activity [10,11], a better understanding of the role of ncRNAs in the context of MYC metabolism would be of great benefit, opening avenues for discovering synthetic lethality targets of MYC-related functions in cancer.

While some reviews have discussed ncRNAs that regulate MYC expression across different tissues [12,13], this review specifically aims to explore ncRNAs that directly influence MYC’s metabolic functions.

While MYC plays a central role in regulating glycolysis, it is not always the case that any ncRNA affecting MYC stability directly impacts glycolytic activity. The relationship between MYC and metabolism is complex and context-dependent, influenced by factors such as cell-type, metabolic state, and compensatory mechanisms. Additionally, MYC’s effects on glycolysis can be modulated by other pathways, including those involving hypoxia and other oncogenes and tumor suppressors such as p53 and AMPK.

Therefore, this review specifically focuses on ncRNAs that have shown clear evidence of modulating MYC-driven metabolic pathways, specifically the Warburg effect, aiming to examine the mechanisms through which these interactions occur. The review also aims to explore the interactions between various types of ncRNAs, shedding light on how they may collaborate or influence each other’s functions in the regulation of MYC’s metabolic function.

By shedding light on these complex relationships, our goal is to deepen the understanding of MYC’s role in metabolism and highlight how ncRNAs contribute to this process. By narrowing the scope to MYC-related metabolic regulation, this offers a more targeted approach to therapeutic development, potentially leading to the identification of biomarkers and synthetic lethality targets to disrupt MYC-driven metabolic reprogramming in cancer.

## 2. Overview of MYC’s Metabolic Function

MYC’s involvement in glycolysis is considered the hallmark of its metabolic reprogramming in vitro and in vivo [7,8,9]. MYC activates the transcription of most glycolytic enzymes, either directly by binding to classical E-box sequences or indirectly through other mechanisms. It is known to enhance the transcription and thus the expression of key glycolytic enzymes and proteins such as glucose transporter 1 (GLUT1), hexokinase 2 (HK2), lactate dehydrogenase A (LDHA), and alpha enolase (Eno1) [7,8,9], a feature that is also observed in MYC^high^ patient samples [14] (Figure 1).

The further direct link between glycolysis and MYC was confirmed in vivo in inducible MYC-driven malignancies analyzed by hyperpolarized 13C pyruvate Magnetic Resonance Spectroscopic Imaging, in which an increase flux of pyruvate to lactate is seen in primary tumors compared to normal or pre-tumor nodules and which was also reversible upon MYC inhibition [15,16]. In addition, further supporting glycolysis, MYC was shown to increase the expression of monocarboxylate transporters MCT1 and MCT2, which are known to be overexpressed in tumors and can mediate lactate efflux, thus maintaining glycolytic rates [17,18] (Figure 1). This was further confirmed in vivo in an Eμ-Myc transgenic mouse model of human B lymphoma, in addition to various MYC-driven malignancies in human data sets [17]. MYC’s influence on pyruvate kinase isoform 2 (PKM2), the enzyme responsible for the final step of glycolysis and which in contrast to PKM1 favors glycolysis over oxidative phosphorylation, was also observed. MYC was found to activate the transcription of splicing factors, promoting the production of PKM2 [19]. Another suggested mechanism was through the activation of the nuclear RNA helicase MTR4 transcription by MYC in hepatocellular carcinoma (HCC) [20]. MTR4 plays a significant role in RNA metabolism and stability and was found to significantly activate glycolytic enzymes expression mainly through alternative splicing regulation in this study. Further confirming the role of MYC in glycolysis were the results from a small-cell lung carcinoma (SCLC) xenograft model, where it has been shown that MYC^high^ tumors are more sensitive to glycolysis inhibition [14].

Despite the extensive knowledge we have about MYC’s metabolic effects, there are still significant gaps in our understanding of the factors that influence these effects. MYC’s impact on metabolism is also context-dependent as it is influenced by various interactions and complex networks that vary across different cell types and conditions.

## 3. Non-Coding RNA and MYC

### 3.1. Overview of ncRNA

Non-coding RNAs (ncRNAs) are diverse RNA molecules that do not encode proteins but regulate gene expression and various cellular processes. They play a crucial role in gene regulatory networks and can directly interact with other RNA molecules. Non-coding RNAs exhibit both oncogenic and tumor-suppressive functions in cancer, and their dysregulation can influence processes such as cell proliferation, apoptosis, invasion, metastasis, and metabolism. Various reviews have extensively studied ncRNAs, describing their different types, mechanisms of action, roles in cancer, and therapeutic potential [21,22,23,24], which we will not delve into; instead, we will provide a brief summary of the key types of ncRNAs and their general functions.

Among the most studied types of ncRNA are long non-coding RNAs (lncRNAs), circular RNAs (circRNAs), and microRNAs (miRNAs), each playing significant roles in cancer progression and cellular metabolism.

LncRNAs are a heterogeneous class of transcripts greater than 500 nucleotides in length and are involved in chromatin remodeling, transcriptional, post-transcriptional, translational, and post-translational regulation. They can function at the transcriptional level by binding to DNA regulatory elements, affecting chromatin remodeling and DNA methylation, which in turn can activate or repress gene transcription. Additionally, lncRNAs play a role at the post-transcriptional level by regulating mRNA stability. One significant mechanism is the “sponging” effect, in which lncRNAs physically bind to and sequester target molecules, such as miRNAs or proteins, preventing them from interacting with their other molecular partners [21]. In addition, lncRNAs can act as scaffolds for RNA-binding proteins, thus affecting RNA processing. Overall, lncRNAs play a crucial role in cancer development by influencing key cellular processes such as cell cycle regulation, proliferation, apoptosis, and migration [21,22].

CircRNAs are formed through back-splicing of pre-mRNA, creating a closed loop that covalently links the 3′ and 5′ ends. These circRNAs play crucial roles in cancer biology, acting as either oncogenes or tumor suppressors, and are highly stable molecules [21]. They are increasingly recognized for their potential as biomarkers and regulators of gene expression. Similar to lncRNAs, circRNAs function as competitive endogenous RNAs (ceRNAs) by containing binding sites for miRNAs, thus facilitating miRNA sponging. By sequestering miRNAs and preventing them from binding to their target mRNAs, circRNAs can increase the stability and expression of these target mRNAs. However, circRNAs can also modulate gene transcription and protein interactions [21,22].

MiRNAs, small RNAs of 20–24 nucleotides, regulate gene expression by binding to mRNA and inhibiting translation or promoting degradation. Dysregulated miRNAs are common in cancers, and they can either function as oncogenes or tumor suppressors. They can influence various cellular processes, including cell proliferation, apoptosis, metastasis, and drug resistance [21,22]. As a result, miRNAs are being explored as potential diagnostic biomarkers, prognostic indicators, and therapeutic targets in cancer. The main mechanism of action entails the miRNA-RISC complex, preventing ribosome binding, thereby inhibiting translation and halting protein synthesis.

Overall, ncRNAs are involved in intricate and complex networks that play crucial roles in regulating various cellular processes. These networks often include feedforward loops, which can help balance the expression of key genes. The function of ncRNAs can be tissue- and context-dependent; the same ncRNA gene may act as a tumor suppressor in one cancer type while promoting disease progression in another, highlighting the complexity and versatility of their functions.

### 3.2. ncRNA and MYC Regulation

Similar to other oncogenes, MYC is intricately linked to non-coding RNAs (ncRNAs) and their complex regulatory networks. On one hand, ncRNAs can regulate MYC expression through a variety of mechanisms. On the other hand, MYC can directly or indirectly influence the expression of ncRNAs, contributing to the dynamic feedback loops that regulate cellular processes such as proliferation, differentiation, and apoptosis. These interactions can either promote or inhibit MYC-driven tumorigenesis, highlighting the complex regulatory network involving ncRNAs and MYC.

LncRNAs can influence MYC mRNA and protein stability in various ways. For example, through miRNA sponging, the lncRNA MYC inhibitory factor (LncRNA-MIF) modulates the expression of F-box and WD repeat domain-containing 7 (FBXW7), leading to MYC protein degradation [25]. On the other hand, GHET1 enhances the binding of MYC mRNA to insulin-like growth factor mRNA-binding proteins (IGFBPs), which regulate translation and mRNA stability [26]. Additionally, MYC has been found to both induce and repress the expression of certain lncRNAs, either directly or indirectly and in a cell-specific manner. This modulation thereby influences gene expression and cellular functions that promote various tumor hallmarks. One example is MINCR (MYC-induced long non-coding RNA), which is overexpressed in several tumors and promotes cell proliferation and Wnt pathway signaling in these tumors [27,28].

Several miRNAs have been found to directly regulate MYC, with the majority doing so through direct binding to the MYC transcript in a canonical fashion [21,29]. However, some miRNAs also exert their effects through indirect mechanisms. For example, miR-24-3p was shown influence MYC protein levels indirectly by targeting OGT, which in turn O-GlcNAcylates the MYC protein, increasing its stability [30]. Most miRNAs that directly bind to MYC mRNA are reduced in cancer, such as the let-7 family [31] and miR-145 [32]. MiRNAs that regulate MYC can also be induced or repressed by MYC itself, leading to the formation of feedback loops. For example, the MYC-induced miR-7-5p can indirectly enhance MYC stability, inducing a positive feedback loop to enhance its own transcription [33], while Let-7a-5p is an example of a MYC-repressed miRNA that directly targets MYC [29]. In addition, several miRNAs were shown to be induced and repressed by MYC, thus mediating a range of MYC tumorigenic functions [34,35].

CircRNAs have also been closely associated with MYC. MYC-driven circRNAs were found to be upregulated in tumors such as triple-negative breast cancer (TNBC) and SCLC and can promote tumor progression in these cells [36,37]. CircRNAs can also act as miRNAs sponges modulating gene expression, including MYC itself [38].

This reciprocal regulation highlights the critical role of ncRNAs in MYC-driven oncogenesis and opens the door to exploring the potential therapeutic effects of these ncRNAs in MYC-driven malignancies.

## 4. Interaction Between ncRNAs and MYC in Cancer Metabolism

In a similar manner, ncRNAs influence MYC-mediated metabolic regulation through three key mechanisms (Figure 2): (1) ncRNAs regulate MYC stability, affecting its degradation or accumulation and thus its ability to control metabolic pathways; (2) ncRNAs with direct metabolic functions are targets of MYC. MYC controls the expression of certain ncRNAs, which can modulate metabolic pathways, such as glycolysis and oxidative phosphorylation; (3) ncRNAs that are targets of MYC regulate MYC expression in a feedback loop, where MYC influences the expression of ncRNAs, while these ncRNAs also affect MYC’s stability and activity, creating a dynamic regulatory cycle.

The following sections describe in detail the ncRNA involved in these mechanisms which have been shown to modulate MYC’s metabolic functions, specifically its glycolytic effect. A list of the ncRNAs described in this review can be found in Table 1.

### 4.1. ncRNAs That Affect Metabolism by Regulating MYC Stability

The role of lncRNA in glucose metabolism through mechanisms linked to MYC has been introduced recently for several tumors. One mechanism is their interaction with IGFBPs (Figure 3). N6-methyladenosine (m^6^A) is a posttranscriptional RNA modification which plays a role in mRNA metabolism, splicing, stability, and translation [58]. IGFBPs have been recently identified as N^6^-methyladenosine (m^6^A) readers that can maintain the stability of m^6^A-modified mRNAs [59] and increase the translation of target transcripts such as MYC [60,61,62]. In colorectal cancer (CRC) cell lines HCT116 and DLD1, Long Intergenic Non-Coding RNA for IGFBP2 Stability (LINIS) was found to directly bind to IGF2BP2, protecting it from autophagic degradation [39]. Knockdown of LINIS in these cell lines was associated with reduced mRNA levels of key oncogenes such as MYC, GLUT1, PKM2, and LDHA, leading to an overall reduction in glycolytic activity. This was rescued to some extent with overexpression of IGF2BP2, indicating a role for LINRIS-IGF2BP2-MYC in glycolysis and cancer proliferation [39]. A correlation between the levels of LINIS and these glycolytic enzymes was also seen in CRC tissue [39]. On the other hand, a tumor suppressor function was seen with LINC00261 in pancreatic cancer [41]. LINC00261 was found to be significantly reduced in pancreatic cancer patient tissue. Furthermore, analysis using a Seahorse analyzer demonstrated that overexpression of LINC00261 decreased both the extracellular acidification rate (ECAR) and oxygen consumption rate (OCR), along with glucose consumption and lactate production, indicating an overall decrease in glycolysis. Additional analysis revealed that LINC00261 reduces MYC activity through two mechanisms. First, it can reduce MYC mRNA stability and expression by sequestering IGF2BP1. Second, LINC00261 acted as a ceRNA, sponging miR-222-3p, thus activating its target Homeodomain Interacting Protein Kinase 2 (HIPK2), which has been shown previously to attenuate MYC activity [41].

It is important to note that other lncRNAs, such as gastric carcinoma high expressed transcript 1 (lncRNA GHET1), were found to enhance the interaction between MYC mRNA and IGF2BP1, thus increasing MYC stability; however, a direct contribution to MYC’s metabolic function was not confirmed [26].

Emphasizing the role of circRNAs in MYC glycolytic function, a study was conducted in prostate cancer (PCa) [42]. circARHGAP29, a circRNA generated from the circularization of the Rho GTPase Activating Protein 29 (*ARHGAP29)* gene, promoted docetaxel resistance in PCa through promoting aerobic glycolysis. Interestingly, circARHGAP29 was found to stabilize LDHA mRNA through enhancing its interaction with IGF2BP2. Interestingly, this circRNA was found to also directly bind to MYC, thus stabilizing MYC mRNA and protein levels, which was further associated with LDHA induction. Inhibition of circARHGAP29 decreased LDHA expression and extracellular glucose consumption in vitro and in vivo, which could be rescued by MYC overexpression.

Another lncRNA which was shown to stabilize MYC is the lncRNA GLCC1, a glycolysis-associated lnRNA which was found to be overexpressed in CRC tissue compared to adjacent tissue and correlated with tumor progression [43]. Through knockdown and in-gain of function expression of GLCC1 in vitro, GLCC1 expression was induced by glucose starvation and resulted in glycolysis, while downregulation of GLCC1 was associated with a reduced extracellular acidification rate (ECAR) and extracellular lactic acid production in various CRC cell lines. Interestingly, GLCC1-induced glycolytic metabolism appeared to be mediated by its direct binding with HSP90, which was significantly increased in glucose starvation conditions. HSP90 protein chaperone is essential for the stability of many proteins and mediates MYC stability in CRC cells [43]. To further confirm the effect of GLCC1, knockdown of GLCC1 was associated with a decrease in MYC expression but not other HSP90 target proteins such as hypoxia-induced factor 1 alpha (HIF1α) or P53. Thus, GLCC1 stabilizes MYC through its direct interaction with HSP90.

CircPDK1 was found to be highly expressed in hypoxic-induced exosomes in pancreatic cancer and was associated with advanced pathology and poor prognosis [44]. In a recent study, circPDK1 played a significant role in tumor growth and in promoting glycolysis in vitro and in vivo in a pancreatic subcutaneous tumor mouse model through a MYC-related pathway [44]. circPDK1 served as a ceRNA sponging miR-628-3p, thereby releasing Bromodomain and PHD finger-containing transcription factor (BPTF) from the inhibiting effects of miR-628-3p. BPTF proteins can regulate the expression of various oncogenes and tumor suppressors through their chromatin-modulating activity. Notably, BPTF is essential for MYC binding of target genes and its overall transcriptional activity [63]. Thus, the study provides insight into the role of hypoxia and HIF1α in inducing circRNA, which can promote the MYC-mediated glycolytic effect [44].

Using castration-resistant prostate cancer cells, miR-644a was found to be a potent tumor suppressor that can inhibit glycolytic activity and the expression of the key glycolytic enzyme glyceraldehyde 3-phosphate dehydrogenase (GAPDH) in vitro and in xenograft models [46]. Interestingly, MYC was found to be a direct target of miR-644a, suggesting the possible mechanism by which miR-644 suppresses the Warburg effect. However, given the broad range of effects of miR-644a downregulating various driver molecules [46], further experiments might be required to establish the exact mechanism. In miR-155ko/ko isolated from a miR-155-deficient breast cancer mouse model, the loss of miR-155 was associated with a decrease in the RNA and protein levels of glucose transporters GLUT1, 3, and 4 and main glycolytic enzymes HK2, PKM2, and LDHA, which was associated with a decrease in MYC expression [48]. Although MYC was not a direct target of miR-155, miR-155 was found to directly and indirectly interact with MYC through its target FOXO3a, which is known to destabilize MYC. MiR-155 was found to indirectly decrease FOXO3 by directly targeting PIK3R1, thereby activating the PI3K-AKT signaling pathway [48]. This activation leads to the phosphorylation of FOXO3, promoting its degradation and therefore increasing MYC. MiR-155’s metabolic effect was further confirmed in human breast cancer cell lines; in addition, miR-155-high tumors displayed higher glucose uptake in patient samples seen through PET images, with miR-155 correlating negatively and positively with FOXO3a and MYC, respectively.

### 4.2. ncRNAs with Direct Metabolic Functions and Regulated by MYC

In addition to their role in stabilizing MYC, another important mechanism by which ncRNAs influence MYC metabolism is through their direct regulation of glycolysis. These ncRNAs are transcriptional targets of MYC and can modulate key metabolic pathways that MYC is involved in, further shaping the cellular metabolic landscape.

The pan-cancer lncRNA Motor neuron and pancreas homeobox 1-antisense RNA1 (MNX1-AS1), which acts as an oncogene promoting the Warburg effect, was found to be a direct target of MYC in [50]. The lncRNA was found to be induced by MYC following epidermal growth factor signaling and facilitates the nuclear transportation of PKM2 by making use of its non-glycolytic function as a coactivator of the transcription of LDHA, PDK1, and GLUT1 genes. Importantly, using MYC-depleted HepG2, silencing of MNX1-AS1 decreased the RNA and protein levels of glycolytic enzymes, which could not be rescued by ectopic expression of MYC, indicating that this could be an essential mechanism by which MYC can mediate its metabolic function. The correlation between MNX1-AS1, nuclear PKM2, and C-MYC was further confirmed in patient tumor samples [50].

MYC can also regulate glycolysis through repressing the expression of ncRNA. MYC, for example, was shown to mediate the repression of the lncRNA IDH1 antisense RNA1 in various cell lines [51]. lncRNA IDH1-AS1 affected MYC–glycolytic function since overexpression of this lncRNA partially inhibited the increase in glucose uptake and lactate production associated with MYC overexpression, while silencing lncRNA IDH1 showed opposite effects. The mechanism of action of this lncRNA was linked to an isocitrate dehydrogenase 1-alpha ketoglutarate–hypoxia-inducible factor 1α (IDH1-α-KG-HIF1α) axis under normoxic conditions only. The lncRNA increases the activity of IDH1, leading to an increase in α-KG, which can suppress HIF1a, limiting its glycolytic function, thus demonstrating an example of how lncRNA can link MYC and HIF1α functions via the Warburg effect.

### 4.3. ncRNAs That Regulate MYC and Are Targets of MYC

Another mechanism that links both previous mechanisms is MYC’s ability to create feedback loops with ncRNA molecules that can induce or suppress glycolysis [52,54,56]. These ncRNAs not only impact MYC stability but are also direct transcriptional targets of MYC, thus creating a dynamic feedback loop by which MYC can sustain its metabolic activity through a self-perpetuating positive feedback mechanism. However, in some cases, these ncRNAs were shown to possess other MYC-independent metabolic regulatory functions that can enhance MYC’s metabolic effect. An example is the sophisticated work carried out with the novel lncRNA glycoLINC (gLINC) induced by MYC [52]. gLINC played a crucial role in forming a complex network with enzymes involved in the glycolytic pathway, including Phosphoglycerate Kinase 1 (PGK1), Phosphoglycerate Mutase 1 (PGAM1), ENO1, PKM2, and LDHA. The assembled complex, known as a metabolon, enhances glycolysis, increases ATP production, and allows cancer cells to survive during serine deprivation in vitro and in vivo. Interestingly, lncRNA promoted MYC expression, suggesting a positive feedback loop. It was also shown that gLINC silencing, although partial, was able to reverse MYC’s glycolytic overexpression effect, indicating that gLINC can enhance MYC-mediated glycolysis. The MYC-gLINC axis therefore shows a mechanism by which cancer cells sustain cell survival during serine deprivation through enhancing glycolysis and sustaining ATP production.

Further support of the importance of m^6^A modifications in cancer metabolism and the crosstalk between m^6^A modification and lncRNA was seen with LncRNA FGF13-AS1 in breast cancer, in which it exhibited a tumor suppressor function reducing MYC stability by directly disrupting MYC-IGF2BP2 binding. Interestingly, the lncRNA was able to attenuate glycolysis cell proliferation, migration, and stemness in these cells; however, it itself was transcriptionally inhibited by MYC [53]. Another study showed the reciprocal regulation between the FTO intronic transcript 1 lncRNA (FTO-IT1) and MYC [54]. Through stabilizing its gene product, the FTO demethylase, this lncRNA was associated with increasing glycolysis in HCC cells and an increase in GLUT1 and PKM2 stability through FTO-mediated demethylation of m^6^A. The expression of this lncRNA was increased by MYC in hypoglycemic conditions; on the other hand, MYC was itself regulated by FTO via m^6^A modification. FTO-IT1 was also associated with tumor growth and glycolysis in a subcutaneous tumor model [54]. These results differ from previous studies that have shown m^6^A modification to be important for the stability of MYC, indicating that m^6^A modification can have context-dependent effects on MYC stability.

Another example was LINC01123, which was upregulated in non-small-cell lung cancer (NSCLC) patients with high ^18^F-FDG uptake on PET/CT scans and correlated with poor survival [56]. A novel function was introduced for this lncRNA with its ability to increase ^18^F-FDG uptake, lactate production, and expression of glycolytic enzymes in vitro and in vivo in a xenograft model. MYC regulated the transcription of LINC01123; interestingly, however, LINC01123 was also found to increase MYC expression through competitively decoying miR-199a-5p, which can bind to MYC and have an inhibitory effect. It is worth noting that studies have also identified miR-199a-5p as a tumor suppressor across various cancer types, with evidence suggesting it can inhibit the Warburg effect by targeting other molecules such as HIF1α [64].

Another interesting ncRNA is lncRNA Myc inhibitory factor (LncRNA-MIF), which is identified as a direct transcript of MYC but has been shown to regulate MYC protein stability in various cell lines, including HeLa and HCT116 [25]. It functions as a ceRNA for miR-586, which modulates the expression of FBWX7, a member of the F-box protein family, acting as a substrate recognition component of an E3 ubiquitin ligase involved in MYC degradation (Figure 4). Consequently, lncRNA-MIF is associated with reduced MYC levels. Knockdown of lncRNA-MIF increases the expression of MYC target genes related to glycolysis, such as GLUT1, LDHA, PKM2, and HK2, while over-expression of the lncRNA was associated with a decrease in the expression of those genes and a decrease in lactate production, confirming its role in regulating glycolysis [25].

Other studies have showed the importance of FBXW7 in decreasing glycolysis and tumor progression in vitro and in vivo [65], underscoring the importance of exploring this axis. The complexity further lies in how FBXW7 in tumors is regulated through various means at the transcriptional and epigenetic level. For example, MYC itself was found to negatively or positively regulate FBXW7 expression through miRNA and lncRNA expression [66].

Overall, these studies show that ncRNAs not only mediate MYC stability, therefore regulating its metabolic function, but also that MYC can regulate the expression of ncRNA, which in turn has metabolic functions, creating a complex regulatory network.

## 5. Clinical Relevance of the ncRNAs Discussed

The majority of the described ncRNAs are still in the early stages of research, with much yet to be explored in terms of their biological functions and mechanisms and clinical significance. However, we provide an overview of the current understanding of their clinical correlations, focusing on their prognostic value and therapeutic potential. We also explore potential avenues for further research into how ncRNAs could be leveraged for MYC-targeted therapies, particularly in the context of synthetic lethality, offering new strategies for selectively targeting MYC-driven cancers.

### 5.1. Prognostic Value and Therapeutic Potential

In a cohort of 95 CRC cases, LncRNA GLCC1 expression was found to be positively correlated with tumor size and invasion depth. Higher levels of GLCC1 expression were associated with more aggressive tumor characteristics and shorter survival times. Univariate and multivariate regression analyses showed that GLCC1 expression was an independent predictor of poor clinical outcomes, indicating its potential use as a prognostic marker [43]. Similarly, LINRIS expression was associated with predicting poor survival in a sample of 62 NSCLC patients [40].

CircPDK1 expression also significantly correlated with tumor size, TNM stage, tumor recurrence, and paclitaxel resistance in 40 patients of NSLC [45].

The low expression of miR-644a, on the other hand, was found to correlate with TNM stage, metastasis, and low overall survival rate in 80 patients with NSCLS [47]. Interestingly, and further demonstrating the complex network of ncRNA interactions, the circRNA CircGLIS3 was shown to function as an oncogene in this study via sponging multiple tumor-suppressive miRNAs, including miR-644a [47]. 

Among miRNA-based therapies, miR-155 inhibition has emerged as one of the advanced methods in clinical development. Cobomarsen is a miR-155 inhibitor designed as a locked nucleic acid (LNA)-modified oligonucleotide [67]. It was tested in a phase 1 clinical trial for conditions such as cutaneous T-cell lymphoma (CTCL), chronic lymphocytic leukemia (CLL), and diffuse large B-cell lymphoma (DLBCL) which are known to be MYC-driven in a number of cases, and adult T-cell leukemia/lymphoma (ATLL) [67]. Following promising results, a phase 2 trial for CTCL was initiated; however, the phase 2 trial was discontinued early due to business decisions [67]. However, with its demonstrated safety and preliminary efficacy, this inhibitor could be tested further in future trials.

The clinical impact of LncRNA MNX1-AS1 is being thoroughly explored and has attracted growing interest. Aberrant expression of MNX1-AS1 has been shown in various tumors, including but not limited to gastric cancer, lung, osteosarcoma, HCC, and ovarian, with its upregulation associated with clinicopathological parameters such as lymphatic metastasis, tumor size, stage, and poor survival [68].

FTO-IT1 was shown to be upregulated in HCC samples and PCa and correlated with decreased overall survival and disease-free survival [54,55]. Similarly, LINC01123 was found to be upregulated and associated with poor prognosis in CRC [57], NSLC [56], and oral squamous cell carcinoma [69].

### 5.2. Clinical Relevance to MYC Regulation and Synthetic Lethality

Given that MYC is aberrantly expressed in nearly 70% of human tumors and is associated with tumor progression, it has long been an attractive therapeutic target [70,71]. However, it remains considered an “undruggable” target due to its pivotal role in normal cell proliferation and its structural characteristics. Specifically, MYC lacks a catalytic cleft, which prevents it from having binding sites for traditional drug modalities, making direct targeting challenging [70]. Alternative approaches for targeting MYC are gradually showing promise and gaining success. These strategies aim to either reduce MYC transcription, translation, or stability; target MYC’s interactions with its binding partners; block MYC’s access to downstream genes; or exploit the vulnerabilities of MYC-dependent tumor cells through efficient synthetic lethality combinations [70,71].

For example, given its role as described in this review for MYC stability, IGF2BP is a promising target for alternative MYC inhibition in cancer treatment. One such strategy involved the use of a small molecule inhibitor of IGF2BP1, BTYNB, which destabilizes MYC mRNA in the melanoma cell line [72]. Given that ncRNAs can regulate IGF2BP, they present exciting therapeutic opportunities. An example of potential therapeutic exploration could involve further investigation into the tumor suppressor lncRNA LINC00261 and its regulation of miR-222-3p. LINC00261 has been shown to be epigenetically regulated, suggesting it could be targeted by epigenetic therapies such as 5-azacytidine, which may help restore its expression [41,73]. While miR-222-3p inhibitors are not yet available for clinical use, they are being studied in several contexts [74]. Another important pathway is linked to the FBXW7 axis. Aurora A kinase, which plays a role in mitotic checkpoints, was found to bind to MYC and inhibit its ubiquitination [71,75]. In accordance with these findings, the specific Aurora A kinase (AURKA) inhibitor MLN8237 (alisertib) has been shown to promote the degradation of MYC through the FBXW7-associated ubiquitin ligase complex. This process induces tumor regression in preclinical models [75], and indeed alisertib has been tested in clinical trials, yielding promising results as a monotherapy [76]. This clearly opens up opportunities to explore the role of lncRNA-MIF [77]. As previously described, enhancing the expression of lncRNA-MIF can promote the degradation of MYC by increasing the expression of FBXW7, making lncRNA-MIF an intriguing target for further investigation. Simultaneously targeting ncRNAs and MYC can potentially enhance the therapeutic effects of AURKA inhibitors while allowing for a reduction in drug dosage, which helps minimize side effects and prevent the development of drug resistance.

Finaly, given that MYC induces a glycolytic phenotype in cancer cells, exploring the synthetic lethality of ncRNAs that directly affect metabolism is a promising approach. This strategy could specifically target cancer cells that are dependent on glycolysis, while sparing normal cells. This idea is supported by evidence from MYC-driven models where inhibition of glycolysis with LDHA inhibitors or inhibitors of the NAD^+^ salvage enzyme nicotinamide phosphoribosyl-transferase (NAMPT) led to selective toxicity in MYC-overexpressing pancreatic cancer and glioblastoma cells [78,79]. Such findings highlight the potential for exploring ncRNA-based therapeutics for ncRNA that have a direct metabolic effect, such as MNXAS-1, to exploit metabolic vulnerabilities in MYC-driven cancers.

With recent advances in MYC targeting, combining MYC inhibitors with ncRNA therapeutics could provide a powerful new strategy for treating MYC-driven cancers.

## 6. Conclusions

In summary, ncRNAs can regulate MYC’s metabolic and glycolytic activity. These RNA classes are known to interact together and create complex networks that are tissue-specific and context-specific. Currently, these studied ncRNAs can affect MYC’s metabolic function by affecting MYC stability, by being a target of MYC but having metabolic functions, or through both mechanisms combined. One of the limitations is that the studied ncRNAs are still in their early research phases, and much remains unknown about their full roles and mechanisms. As further studies are conducted, it is likely that new mechanisms will emerge, such as some ncRNAs currently classified in the second category potentially being reclassified into the third category or entirely new mechanisms being discovered to explain their functions. Another potential limitation is that findings from in vitro studies may not always fully reflect the in vivo mechanisms, particularly in the context of MYC-mediated metabolic regulation. The complexity of MYC’s role in metabolism is influenced by a variety of external and internal factors, such as hypoxia, and cell-specific conditions, which are difficult to replicate in simplified in vitro models. These additional factors can significantly alter MYC’s impact on metabolism and MYC’s interaction with ncRNA. It is therefore crucial to investigate the role of these ncRNAs in vivo and in patient samples to better understand their true impact on MYC-driven metabolic regulation. In addition, with recent advances in MYC targeting, combining MYC inhibitors with ncRNA therapeutics could provide a powerful new strategy for treating MYC-driven cancers.

## Figures and Tables

**Figure 1 ncrna-11-00027-f001:**
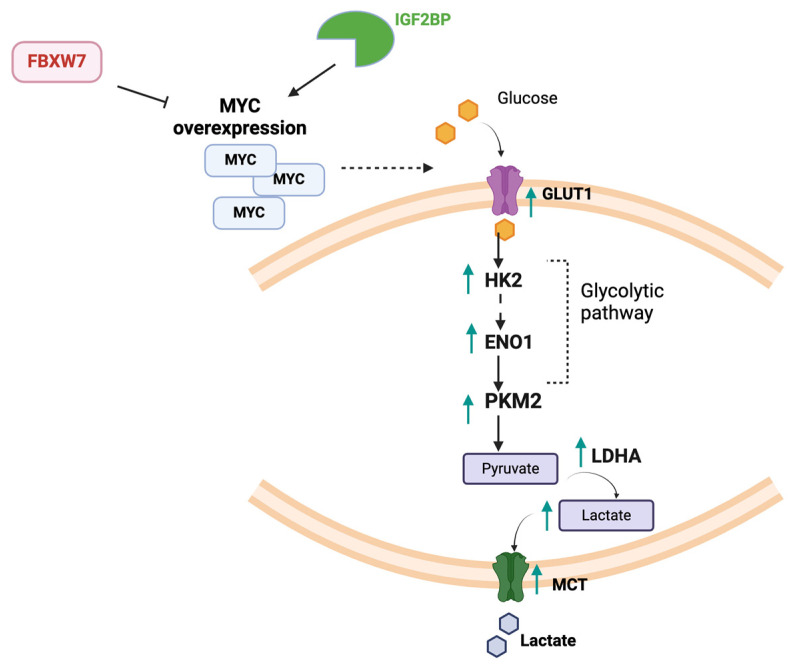
A schematic illustration of MYC’s glycolytic effect. This diagram depicts how MYC overexpression influences glycolysis, beginning with an increase in GLUT1 expression to enhance glucose uptake into cells. MYC also upregulates the transcription of key glycolytic enzymes, such as HK2, ENO1, PKM2, and finally LDHA, which is responsible for converting pyruvate to lactate. Additionally, MYC enhances the expression of MCT transporters, promoting lactate efflux from the cells. FBXW7 promotes MYC degradation, reducing its levels, while IGF2BP stabilizes MYC mRNA, promoting its translation.

**Figure 2 ncrna-11-00027-f002:**
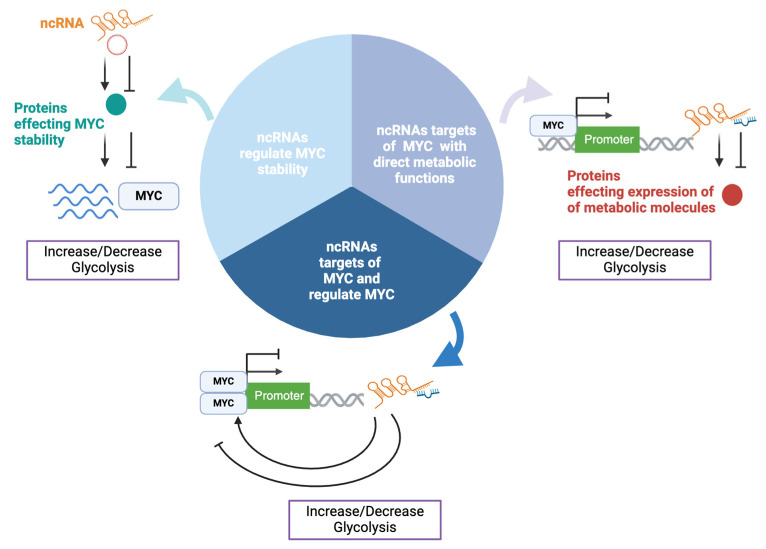
A schematic illustration of mechanisms of ncRNA regulation of MYC and metabolism. This diagram illustrates three distinct mechanisms by which ncRNAs regulate MYC and its metabolic activity: (1) ncRNAs modulate MYC stability, (2) MYC can regulate the expression of ncRNAs with direct roles in metabolism, and (3) MYC-regulated ncRNAs affect MYC expression.

**Figure 3 ncrna-11-00027-f003:**
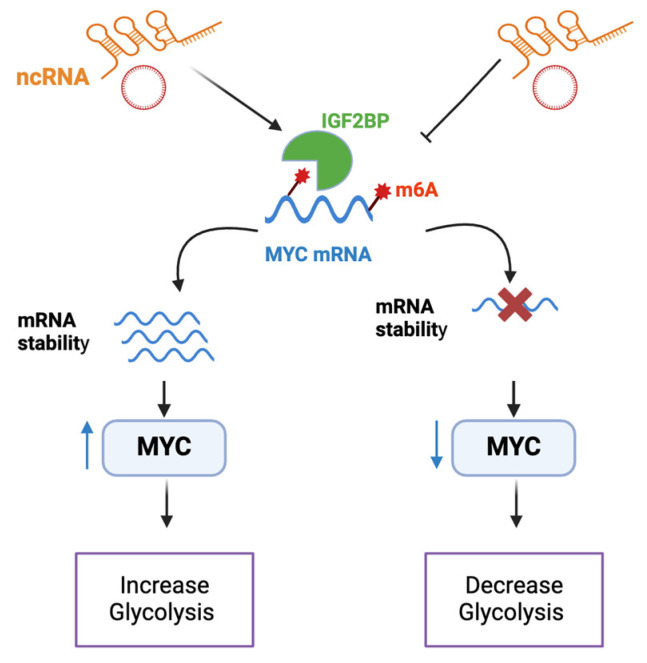
A schematic illustration of ncRNA regulating IGF2BP-MYC axis to control glycolysis. ncRNA can modulate the stability of IGF2BP or its binding to MYC mRNA. This regulation ultimately affects IGF2BP’s ability to stabilize MYC mRNA, leading to either an increase or decrease in MYC levels. Consequently, MYC glycolytic activity is influenced, resulting in an upregulation or downregulation of glycolysis.

**Figure 4 ncrna-11-00027-f004:**
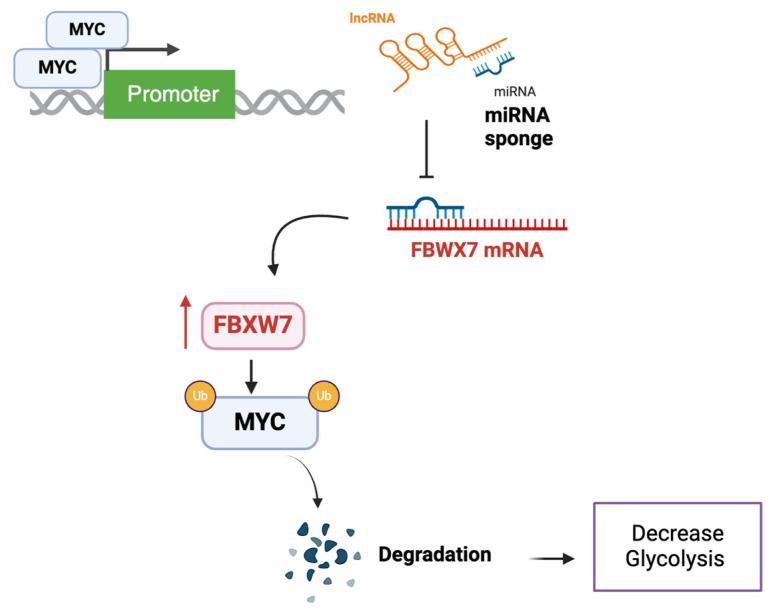
A schematic illustration of lncRNA-mediated sponging of miRNAs regulating FBWX7-MYC axis to control glycolysis. MYC, as a transcription factor, drives the expression of a specific lncRNA, which in turn acts as a miRNA sponge. By sequestering miRNAs, the lncRNA relieves their inhibitory effect on FBXW7, leading to its stabilization and subsequent degradation of MYC. This results in a reduction in MYC levels and glycolytic activity.

**Table 1 ncrna-11-00027-t001:** List of ncRNAs involved in MYC-mediated glycolytic effect.

ncRNA	Relation to MYC	Effect on Metabolism/Tumorigenesis	Clinical Relevance	Refs.
ncRNAs that affect metabolism by regulating MYC Stability				
LINRIS	Stabilizes MYC via IGFBP2 interaction	Increases glycolysis and promotes cancer proliferation	Poor prognosis in NSCLC patients	[39,40]
LINC00261	Reduces MYC via IGF2BP2 interaction and miRNA sponging	Decreases glycolysis	Decreased—poor prognosis in Pancreatic cancer	[41]
circARHGAP2	Stabilizes MYC and LDHA via IGFBP2 interaction	Increases glycolysis and promotes docetaxel resistance		[42]
GLCC1	Stabilizes MYC via HSP90 interaction	Increases glycolysis and correlates with CRC progression	Poor prognosis in CRC	[43]
circPDK1	Increases MYC activity via miRNA sponging	Increases glycolysis and promotes cancer proliferation	Poor prognosis in pancreatic cancer	[44,45]
miR-644a	Inhibits MYC directly	Decreases glycolysis	Poor prognosis in NSCLC patients	[46,47]
miR-155	Inhibits MYC indirectly via FOXO3a	Decreases glycolysis	Decreased—poor prognosis in breast cancer	[48,49]
ncRNAs with direct metabolic functions and regulated by MYC				
MNX1-AS1	Induced by MYC	Increases glycolysis and cancer progression		[50]
IDH1-AS1	Repressed by MYC	Decreases glycolysis and suppresses HIF1α		[51]
ncRNAs that regulate MYC and are targets of MYC				
gLINC	Induced by MYC and activates MYC	Increases glycolysis and supports cell survival under serine deprivation conditions		[52]
FGF13-AS1	Reduces MYC via IGF2BP2 interaction	Decreases glycolysis and stemness		[53]
FTO-IT1	Induced by MYC and stabilizes MYC via m6A modification	Increases glycolysis	Poor prognosis in HCC and PCa	[54,55]
LINC01123	Induced by MYC and increases MYC expression via miRNA sponging	Increases glycolysis	Poor prognosis in CRC and NSCLC	[56,57]
LncRNA-MIF	Induced by MYC and inhibits MYC via miRNA sponging and FBXW7 interaction	Decreases glycolysis		[25]

## Data Availability

Not applicable.

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
