# Peer review of "The Role of Non-Coding RNAs in MYC-Mediated Metabolic Regulation: Feedback Loops and Interactions"

_ncrna, 2025, doi:10.3390/ncrna11020027_

Round 1
Reviewer 1 Report
Comments and Suggestions for Authors
The three key mechanisms of ncRNA influence (stability, as targets in direct metabolic function and targets that regulate MYC) on MYC-mediated metabolic regulation are well described. What is missing is a description (figure) of MYC regulation of glucose. The reader needs to be able to see directly in this review (not somewhere else) where key proteins/enzymes regulated by MYC are located in the glucose pathway. GLUT1, HK2, LDHA, ENO1, PKM2, MTR4, HIF1a, PGK1, IGFBP and most importantly FBXW7.
Comments on the Quality of English LanguageThere are some areas where grammar could be improved.
line 143- not "introduced" but identified, line 154-"In the same mechanism, however with opposite effects" change to similarly but with opposite effects
line 192- casteration is castration, line 292- This study further confirms other studies that show the change to Studies show....
line 331- is the clincail impact, clinical, line 341 promosing is promising
Author Response
Comment 1:The three key mechanisms of ncRNA influence (stability, as targets in direct metabolic function and targets that regulate MYC) on MYC-mediated metabolic regulation are well described. What is missing is a description (figure )of MYC regulation of glucose .The reader needs to be able to see directly in this review (not somewhere else) where key proteins/enzymes regulated by MYC are located in the glucose pathway. GLUT1, HK2, LDHA, ENO1, PKM2, MTR4, HIF1a, PGK1, IGFBP and most importantly FBXW7
Response 1 :
Thank you for your feedback. I agree that a figure on MYC regulation of glucose metabolism would improve the clarity of its effect. I added a schematic figure 1 (line 103) showing key enzymes and proteins involved in glycolysis that are regulated by MYC in section of Overview on MYC metabolic with a simple demonstartion of how FBXW7 and IGF2BP2 can fit in.
Comment 2:
Comments on the Quality of English Language.There are some areas where grammar could be improved.
Line 143- not "introduced" but identified, line 154-"In the same mechanism, however with opposite effects" change to similarly but with opposite effects
line 192- casteration is castration, line 292- This study further confirms other studies that show the change to Studies show....
line 331- is the clincail impact, clinical, line 341 promosing is promising
Response 2 :
Line 242 described as been changed to identified
Line 299 casteration was changed to castration ,Line 404 “This study further confirms other studies that show the change to Studies show” was changed to “Other studies have showed”
Line 449 clinical has been adjusted, promising has been adjusted
Editing and proof reading was further conducted in the entire document
Reviewer 2 Report
Comments and Suggestions for Authors
The authors here provided a summary of the relationship between non-coding RNAs and MYC in regulating metabolism. The contents are clearly organized and can give much more thoughts about the ncRNAs’ role in MYC regulation. Overall, it is a nice summary for readers. I also have some suggestions for the reference.
1. There are indeed too many ncRNAs involved in the MYC-driven metabolism regulation. Maybe a table including all these categoried ncRNAs’ basic information and biological function can be given.
2. In the abstract, the authors also refer to synthetic lethality. I did not check this keywords in the following texts, but I can find many gene combinations in enhancing MYC-mediated metabolism regulation axis from the examples. It is indeed a interesting topic. I suggest the authors also add this discussion in the part “3. Clinical relevance of the ncRNAs discussed”.
3. If a interaction network of MYC, ncRNAs and their targets can be made with cytospace, it will be more easily accessible to readers.
Author Response
Comment 1:
The authors here provided a summary of the relationship between non-coding RNAs and MYC in regulating metabolism. The contents are clearly organized and can give much more thoughts about the ncRNAs’ role in MYC regulation. Overall, it is a nice summary for readers. I also have some suggestions for the reference.
There are indeed too many ncRNAs involved in the MYC-driven metabolism regulation. Maybe a table including all these categoried ncRNAs’ basic information and biological function can be given.
Response 1:
Thank you for your feedback .A table has been added (Table 1)to include all the ncRNA discussed, The table provides a clear, organized overview of each ncRNA's relationship to MYC, its impact on metabolism, and its clinical relevance, allowing the reader to quickly reference and compare key information. Line 222
Comment 2:
In the abstract, the authors also refer to synthetic lethality. I did not check this keywords in the following texts, but I can find many gene combinations in enhancing MYC-mediated metabolism regulation axis from the examples. It is indeed a interesting topic. I suggest the authors also add this discussion in the part “3. Clinical relevance of the ncRNAs discussed”.
Response 2:
I thank the reviewer for raising this important point. In response, I have added a brief section under Subheading 5.2: Clinical Relevance to MYC Regulation and Synthetic Lethality (line 464), which provides an overview of the current status of MYC targeting. This section also highlights potential areas where ncRNAs could play a role in MYC regulation and synthetic lethality, offering insights into how ncRNAs may contribute to therapeutic strategies targeting MYC-driven cancers. As this is still a relatively new area of research, there is limited literature available, but the potential for ncRNAs to contribute to these therapeutic approaches is an exciting prospect.
Comment 3:
If a interaction network of MYC, ncRNAs and their targets can be made with cytospace, it will be more easily accessible to readers.
Response 3:
I thank the reviewer for the suggestion. While I appreciate the idea, I have presented this information in a table format instead. The table provides a comprehensive summary of all the ncRNAs discussed, along with their links to MYC, metabolism, and clinical relevance. I believe this approach effectively conveys the key points and allows readers to easily access the relevant information in one concise and accessible format. I hope this is sufficent for the reviewer
Reviewer 3 Report
Comments and Suggestions for Authors
The review (ncRNA-3483234) discusses a clear description of the involvement of ncRNAs in MYC-regulated metabolic control, especially the Warburg effect by enhancing glycolysis. The article is very well written and highlights the clinical implications of various ncRNAs, providing an up-to-date summary of their potential roles in cancer prognosis therapy. I have the following concerns.
- The structure of the content needs to be rearranged for easier reading. I feel that switching from the metabolic functions of MYC to the mechanisms of ncRNAs comes across as abrupt. A better ordering, with subheadings and a coherent line of thought, would make it easier for readers to follow the story. The review could also be improved by including a summary table or figure that categorizes the ncRNAs discussed, their action mechanisms, and their clinical value.
- While the review is extensive and wide-ranging in context about ncRNAs and MYC interactions, mechanistic detail is sometimes shallow. Descriptions of how individual ncRNAs (e.g., LINRIS, circARHGAP29) regulate MYC stability or metabolic pathways are brief. Incorporating additional descriptive molecular pathways, such as signaling cascades or post-translational modifications, would strengthen the review. Moreover, the review may be complemented with a more elaborate description of the context-dependent nature of these interactions as MYC's role in metabolism can be significantly different in different types of cancers and microenvironments.
- The review briefly discusses the clinical relevance of ncRNAs in MYC-driven cancers, but this part is underdeveloped.
- While some ncRNAs (e.g., miR-155, MNX1-AS1) are cited about clinical trials or as predictors, the combination of ncRNA therapy with MYC inhibitors is merely addressed.
- Extending the discussion to cover such aspects as drug delivery complications, specificity, and potential side effects would present a more even view of the therapeutic picture. Secondly, the review can be enhanced through the assessment of ongoing clinical trials targeting ncRNAs in MYC-induced malignancies as it would provide insight into the translational value of the research.
Author Response
Comment 1:
The review (ncRNA-3483234) discusses a clear description of the involvement of ncRNAs in MYC-regulated metabolic control, especially the Warburg effect by enhancing glycolysis. The article is very well written and highlights the clinical implications of various ncRNAs, providing an up-to-date summary of their potential roles in cancer prognosis therapy. I have the following concerns.
The structure of the content needs to be rearranged for easier reading. I feel that switching from the metabolic functions of MYC to the mechanisms of ncRNAs comes across as abrupt. A better ordering, with subheadings and a coherent line of thought, would make it easier for readers to follow the story.
Response 1:
Thank you for your feedback , in an attempt to improve the ordering and flow of the content, I have made some modifications.
Section 3 is now titled 3. Non-coding RNA and MYC, with two subheadings:
3.1. Overview of ncRNA: This section provides a general overview of non-coding RNAs (ncRNAs) and their mechanisms of action. It discusses briefly the different types of ncRNAs and their roles in gene regulation, including transcriptional and post-transcriptional control, as well as their involvement in various cellular processes.
3.2. ncRNA and MYC Regulation: This section serves as a transition to show the link between ncRNAs and the regulation of MYC. It briefly explores how ncRNAs directly or indirectly regulate MYC expression and contribute to the intricate networks that control cellular growth and oncogenesis.
While Section 4 titled 4. Interaction between ncRNAs and MYC in Cancer Metabolism: dives into discussing the metabolic aspect
Additionally section 5 title 5. Clinical relevance of the ncRNAs discussed has been divided into two subheading
5.1. Prognostic value and theraputic potential : This section provided the current known prognostic values of the discussed RNA
5.2. Clinical Relevance to MYC regulation and synthetic lethality: This section discusses the current approached for targeting MYC and MYC driven malignancies and provides insight on how ncRNA can fit
Additionally, in line 99 the following was added to show the gap of knowledge and as transition between the over view of MYC metabolic effect and the ncRNA sections "Despite the extensive knowledge we have about MYC's metabolic effects, there are still significant gaps in our understanding of the factors that influence these effects. MYC's impact on metabolism is also context-dependent, as it is influenced by various interactions and complex networks that vary across different cell types and conditions"
Comment 2:
The review could also be improved by including a summary table or figure that categorizes the ncRNAs discussed, their action mechanisms, and their clinical value.
Response 2:
A table has been added to include all the ncRNA discussed, The table provides a clear, organized overview of each ncRNA's relationship to MYC, its impact on metabolism, and its clinical relevance, allowing the reader to quickly reference and compare key information.
Comment 3:
While the review is extensive and wide-ranging in context about ncRNAs and MYC interactions, mechanistic detail is sometimes shallow. Descriptions of how individual ncRNAs (e.g., LINRIS, circARHGAP29) regulate MYC stability or metabolic pathways are brief. Incorporating additional descriptive molecular pathways, such as signaling cascades or post-translational modifications, would strengthen the review. Moreover, the review may be complemented with a more elaborate description of the context-dependent nature of these interactions as MYC's role in metabolism can be significantly different in different types of cancers and microenvironments.
Response 3:
Thank you for your feedback , to address this more mechanistic details was added to the ncRNA that were not discussed thoroughly . Particularly LINRIS (L-244),LINC00261 (L-251),circARHGAP29 (L269),GLCC1 (L-277),mir155 (L-305) all highlighted in yellow
Comment 4:
The review briefly discusses the clinical relevance of ncRNAs in MYC-driven cancers, but this part is underdeveloped.While some ncRNAs (e.g., miR-155, MNX1-AS1) are cited about clinical trials or as predictors, the combination of ncRNA therapy with MYC inhibitors is merely addressed.Extending the discussion to cover such aspects as drug delivery complications, specificity, and potential side effects would present a more even view of the therapeutic picture
Response 4:
I thank the reviewer for raising this important point. In response, I have added a brief section under Subheading 5.2: Clinical Relevance to MYC Regulation and Synthetic Lethality, which provides an overview of the current status of MYC targeting. This section also highlights potential areas where ncRNAs could play a role in MYC regulation and synthetic lethality, offering insights into how ncRNAs may contribute to therapeutic strategies targeting MYC-driven cancers. As this is still a relatively new area of research, there is limited literature available, but the potential for ncRNAs to contribute to these therapeutic approaches is an exciting prospect.
However, I did not want to delve too deeply into aspects such as drug delivery complications, specificity, and potential side effects, as this would extend the scope of the current discussion. These topics are certainly important for a more comprehensive understanding of the therapeutic potential, but they may be more appropriately covered in other reviews that focus more extensively on these areas
Comment 5:
Secondly, the review can be enhanced through the assessment of ongoing clinical trials targeting ncRNAs in MYC-induced malignancies as it would provide insight into the translational value of the research.
Response 5:
Out of the discussed ncRNA, miR-155 inhibitors are the most advanced with clinical trials. Therefore, I have included more detailed information about its current status, as mentioned in line 455.
Line 445
"Among miRNA-based therapies, miR-155 inhibition has emerged as one of the advanced in clinical development. Cobomarsen, is an miR-155 inhibitor designed as a locked nucleic acid (LNA)-modified oligonucleotide [67]. It was tested in a phase 1 clinical trial for conditions such as cutaneous T-cell lymphoma (CTCL), chronic lymphocytic leukemia (CLL), diffuse large B-cell lymphoma (DLBCL) which are known to be MYC-driven in a number of cases, and adult T-cell leukemia/lymphoma (ATLL) [67]. Following promising results, a phase 2 trial (for CTCL was initiated, however, the phase 2 trial was discontinued early due to business decisions[67]. However, with its shown safety and preliminary efficacy this inhibitor could be tested further in future trials."
Round 2
Reviewer 1 Report
Comments and Suggestions for Authors
Nice review!
Reviewer 2 Report
Comments and Suggestions for Authors
Good work. The revised manuscript can be accepted now.
Reviewer 3 Report
Comments and Suggestions for Authors
The article is well-formatted and properly corrected. I recommend it for acceptance.
The overall quality of the English language in the article is good. The writing is clear and mostly coherent, and it is improved for better flow.